# Generation of the Chondroprotective Proteomes by Activating PI3K and TNFα Signaling

**DOI:** 10.3390/cancers14133039

**Published:** 2022-06-21

**Authors:** Xun Sun, Ke-Xin Li, Marxa L. Figueiredo, Chien-Chi Lin, Bai-Yan Li, Hiroki Yokota

**Affiliations:** 1Department of Pharmacology, School of Pharmacy, Harbin Medical University, Harbin 150081, China; sunxun@iu.edu (X.S.); kexli@iu.edu (K.-X.L.); 2Department of Biomedical Engineering, Indiana University Purdue University Indianapolis, Indianapolis, IN 46202, USA; lincc@iupui.edu; 3Department of Basic Medical Sciences and Interdisciplinary Biomedical Sciences Program, Purdue University, West Lafayette, IN 47907, USA; mlfiguei@purdue.edu; 4Indiana Center for Musculoskeletal Health, Indiana University School of Medicine, Indianapolis, IN 46202, USA; 5Simon Cancer Center, Indiana University School of Medicine, Indianapolis, IN 46202, USA

**Keywords:** chondrosarcoma, chondrocytes, MSC, inflammation, TNFα, Hsp90ab1, GAPDH

## Abstract

**Simple Summary:**

Chondrosarcoma and inflammatory arthritis are two joint-damaging diseases. Here, we examined whether a counterintuitive approach of activating tumorigenic and inflammatory signaling may generate joint-protective proteomes in mesenchymal stem cells and chondrocytes for the treatment of chondrosarcoma and inflammatory arthritis. While activating PI3K signaling and the administration of TNFα to chondrosarcoma cells and chondrocytes promoted tumor progression and inflammatory responses, those cells paradoxically generated a chondroprotective conditioned medium. Notably, the chondroprotective conditioned medium was enriched with Hsp90ab1 that interacted with GAPDH. Extracellular GAPDH interacted with L1CAM, an oncogenic transmembrane protein, and inhibited tumorigenic behaviors, whereas intracellular GAPDH downregulated p38 in chondrocytes and exerted anti-inflammatory effects. The result supports the unconventional approach of generating chondroprotective proteomes.

**Abstract:**

Purpose: To develop a novel treatment option for Chondrosarcoma (CS) and inflammatory arthritis, we evaluated a counterintuitive approach of activating tumorigenic and inflammatory signaling for generating joint-protective proteomes. Methods: We employed mesenchymal stem cells and chondrocytes to generate chondroprotective proteomes by activating PI3K signaling and the administration of TNFα. The efficacy of the proteomes was examined using human and mouse cell lines as well as a mouse model of CS. The regulatory mechanism was analyzed using mass spectrometry-based whole-genome proteomics. Results: While tumor progression and inflammatory responses were promoted by activating PI3K signaling and the administration of TNFα to CS cells and chondrocytes, those cells paradoxically generated a chondroprotective conditioned medium (CM). The application of CM downregulated tumorigenic genes in CS cells and TNFα and MMP13 in chondrocytes. Mechanistically, Hsp90ab1 was enriched in the chondroprotective CM, and it immunoprecipitated GAPDH. Extracellular GAPDH interacted with L1CAM and inhibited tumorigenic behaviors, whereas intracellular GAPDH downregulated p38 and exerted anti-inflammatory effects. Conclusions: We demonstrated that the unconventional approach of activating oncogenic and inflammatory signaling can generate chondroprotective proteomes. The role of Hsp90ab1 and GAPDH differed in their locations and they acted as the uncommon protectors of the joint tissue from tumor and inflammatory responses.

## 1. Introduction

In response to chemotherapeutic agents, the reverse behavior of resilient cancer cells is frequently observed. While their tumorigenic action may initially be suppressed, cells tend to develop drug resistance and gradually strengthen their progression [1]. This unintended reaction can be interpreted with the reaction force in mechanics and the law of natural selection. The former tells that any action receives the same magnitude of reaction in the opposite direction, while the latter states that natural selection via cell competition favors cells with the survival of the fittest [2]. By considering these two fundamental laws, we developed an unconventional, yet rational therapeutic strategy for skeletal cancer and long-lasting inflammatory diseases, using chondrosarcoma (CS) and arthritis as a model system. 

Although skeletal cancer and degenerative joint diseases such as osteoarthritis (OA) and rheumatoid arthritis (RA) are independent disorders, some of the underlying processes are tightly coupled [3]. Chronic inflammation is reported to facilitate tumor progression, while the induction of inflammation may contribute to suppressing tumorigenesis by stimulating immune responses [4]. An uncommon question is whether the development of musculoskeletal cancers such as CS and chronic joint diseases such as inflammatory arthritis can be managed by the same therapeutic strategy. Here, we aimed to generate a chondroprotective proteome from mesenchymal stromal cells (MSCs) and chondrocytes and examined the possibility of protecting skeletal tissues from CS and inflammatory arthritis.

CS is the second most common type of bone cancer that originates from cartilage cells, while OA is the most frequent form of arthritis caused by overuse and traumatic injuries to the joints [5,6] and RA is an autoimmune and inflammatory joint disease [7]. Although the exact cause of CS is not known, the standard treatment is surgery coupled with radiation, with only limited data supporting the role of chemotherapy [8]. The treatment option for RA includes the administration of TNFα inhibitors, which interfere with the inflammatory action of TNFα and suppress the immune reaction [9]. Besides RA, OA is also associated with the inflammation and destruction of cartilage tissues [10]. The current OA treatment includes the administration of acetaminophen and nonsteroidal anti-inflammatory drugs for pain relief, physical and occupational therapies to strengthen surrounding muscles, and surgical procedures to lubricate joints and realign bones [11]. These conventional OA treatments may reduce pain with an increase in joint mobility, but joint replacement may eventually be needed since no existing treatments can reverse OA progression.

As a novel therapeutic option for CS and inflammatory arthritis, examined in this study was a counterintuitive approach of activating oncogenic and inflammatory signaling. Although paradoxical, this unorthodox approach was rationalized by the concept of cell competition [12]. In the development of Drosophila’s wing discs, a group of proliferating cells with a high protein synthesis rate is shown to induce the death of neighboring unfit cells [13,14]. In mouse embryos, c-Myc-overexpressing cells are reported to remove neighboring cells with a low level of c-Myc [15,16]. These two examples indicate that the expression of various proteins or transcription factors can be explained based on a “survival of the fittest” principle. Consistently, we have previously shown that the activation of cell-proliferating pathways such as Wnt signaling in osteocytes, MSCs, and even breast cancer cells converted these cells into induced tumor-suppressing cells (iTSCs) [17,18,19]. Notably, iTSC-derived secretomes acted as anti-tumor agents and inhibited the proliferation of tumor cells. The secretomes also suppressed cancer cell migration by downregulating MMPs and prevented bone resorption by decreasing cathepsin K in osteoclasts [17,18,19].

While the inhibition of PI3K/Akt is a common practice in chemotherapy [20], the activation of PI3K signaling by overexpressing Akt in MSCs also converted them into iTSCs [19]. The first intriguing question herein was whether the activation of PI3K signaling in MSCs may generate not only iTSCs but also induced inflammation-suppressive cells (iISCs). The second question was whether iTSCs and iISCs would be generated by treating host cells such as MSCs, chondrocytes, and synovial cells with TNFα, a pro-inflammatory cytokine [21]. While the TNFα-based approach is radically unconventional because of the inflammatory action of TNFα, it is reported that PI3K signaling can promote the production of pro-inflammatory cytokines [22]. Taken together, we hypothesized that although the administration of TNFα to MSCs and joint cells such as chondrocytes and synovial cells induces inflammatory reactions, their CM may act as the anti-inflammatory agent in the same manner as the anti-tumor proteome.

In this study, MSCs, chondrocytes, and synovial cells were employed as three types of cell sources of iTSCs and iISCs, and their CMs were generated by a pharmacological activator of PI3K, as well as TNFα. We evaluated the efficacy of suppressing the progression of CS cells and reducing inflammatory responses and MMP production in chondrocytes and synovial cells. We also examined whether the proteomes in CMs could inhibit the development of bone-resorbing osteoclasts and enhance the synthesis of ECM proteins in the cartilage such as type II collagen and aggrecan.

To investigate the regulatory mechanism underlying the chondroprotection, we first evaluated a group of tumor-suppressing proteins such as heat shock protein 90 beta (Hsp90ab1), calreticulin (Calr), histone H4 (H4), polyubiquitin C (Ubc), and enolase 1 (Eno1), all of which were commonly enriched in iTSC-derived CMs in our previous studies [17,19,23]. Among them, Hsp90ab1 was identified as a prime candidate, at least in part, responsible for the observed chondroprotective action of MSC-derived CM. The immunoprecipitation assay, followed by mass spectrometry-based whole-genome proteomics analyses, revealed that Hsp90ab1 interacted with glyceraldehyde 3-phosphate dehydrogenase (GAPDH). Although GAPDH is often treated as a housekeeping gene [24], a regulatory analysis revealed its novel anti-tumor, anti-inflammatory action by interacting with L1CAM, an oncogenic transmembrane protein [25]. Collectively, this study provided the possibility of generating iTSCs and iISCs from MSCs and utilizing their CM as a therapeutic agent for CS and inflammatory arthritis.

## 2. Materials and Methods

### 2.1. Cell Culture

SW1353 and OUMS-27 human CS cells, C28/I2 chondrocytes (obtained from Dr. M. Goldring, Hospital for Special Surgery, New York, NY, USA), and 4T1.2 mouse mammary tumor cells (obtained from Dr. R. Anderson at Peter MacCallum Cancer Institute, Melbourne, Australia) were cultured in DMEM [26,27,28,29] RAW264.7 pre-osteoclast cells (ATCC, Manassas, VA, USA) were grown in αMEM [30]. Human MSCs (Lonza, Basel, Switzerland) were grown in MSC Basal Medium (Lonza). MH7A synovial cells (Riken Cell Bank, Tsukuba, Japan) were cultured in RPMI1640 [31]. The culture media were supplemented with 10% FBS (fetal bovine serum) and antibiotics (penicillin and streptomycin), and cells were maintained at 37 °C and 5% CO_2_.

### 2.2. Agents

SW1353 human CS cells, MSCs, C28/I2 chondrocytes, and MH7a synovial cells were treated with 50 µM of YS49 (HY-15477, MCE, Monmouth Junction, NJ, USA) or 10 ng/mL of TNFα (NBP2-35076, Novus, CO, USA). C28/I2 chondrocytes were treated with recombinant Hsp90ab1 (OPCA05157; Aviva System Biology, San Diego, CA, USA), Calreticulin (MBS2009125, MyBioSource, San Diego, CA, USA), Histone H4 (MBS2097677; MyBioSource), Ubiquitin C (MBS2029484; MyBioSource), and Enolase 1 (MBS2009113; MyBioSource). SW1353 and OUMS-27 human CS cells were treated with TNFα (10 ng/mL) and GAPDH (2 µg/mL, ab82633, Abcam, Cambridge, UK) recombinant proteins. Of note, the effects of YS49- or TNFα-treated CMs were evaluated compared to the untreated control CMs using the same basal medium such as DMEM, αMEM, MSC basal medium, and RPMI1640.

### 2.3. Preparation of Conditioned Medium (CM) and ELISA Assay

For in vitro experiments, cells were treated with agents for 1 day. The medium was then exchanged to remove agents, and the cells were incubated for 1 additional day with the consistent basial medium. CM was subjected to low-speed centrifugation at 2000 rpm for 10 min. The cell-free supernatants were centrifuged at 4000 rpm for 10 min and subjected to filtration with a 0.22-μm polyethersulfone membrane (Sigma, St. Louis, MO, USA). The supernatants were further centrifuged at 10,000× *g* for 30 min at 4 °C to remove remaining cell debris and at 100,000× *g* (Type 90 Ti Rotor, Beckman, Brea, CA, USA) overnight at 4 °C to remove exosomes. For in vivo experiments, CM was harvested from the fetal bovine serum-free medium and treated by a filter with a cutoff molecular weight of 3 kDa. The levels of aggrecan and type II collagen in C28/I2 chondrocytes in response to 1 µg/mL of Hsp90ab1 were determined using ELISA kits (MBS262707 and MBS263555; MyBioSource).

### 2.4. MTT and EdU Assays

MTT-based metabolic activity was evaluated using ~2000 cells seeded in 96-well plates (3585, Corning, Glendale, AZ, USA). CM and drugs were given on day 2, and cells were dyed with 0.5 mg/mL thiazolyl blue tetrazolium bromide (M5655, Sigma, St. Louis, MO, USA) on day 4 for 4 h. Optical density for assessing metabolic activities was determined at 570 nm using a multi-well spectrophotometer.

Using the EdU procedure previously described [32], cellular proliferation was examined using a fluorescence-based cell proliferation kit (Click-iT™ EdU Alexa Fluor™ 488 Imaging Kit; Thermo-Fisher, Waltham, MA, USA). Approximately 800 cells were seeded in 96-well plates (Corning) on day 1, CMs were added on day 2, and cells were labeled with 10 µM EdU on day 4 for 4 h. After labeling, cells were fixed in a 3.7% (*w*/*v*) formaldehyde solution for 15 min at room temperature. They were washed with a PBS buffer (3% BSA, 0.5% Triton^®^ X-100) and incubated with a freshly prepared Click-iT^®^ reaction cocktail in dark for 30 min at room temperature. After rinsing with a PBS buffer, eight images from four wells in each group were taken with a fluorescence microscope (magnification, 100×, Olympus, Tokyo, Japan). The number of fluorescently labeled cells, as well as the total number of cells, were counted using Image J (National Institutes of Health, Bethesda, MD, USA) and the ratio of the fluorescently labeled cells to the total cells was determined.

### 2.5. Transwell Invasion and Scratch Motility Assays

A transwell invasion assay was conducted to evaluate invasive motility. The invasion capacity of CS cells was determined using a 12-well plate and transwell chambers (Thermo Fisher Scientific, Waltham, MA, USA) with 8-µm pore size. Transwell chambers were coated with 300 µL Matrigel (100 µg/mL) that was polymerized and dried overnight. After adding 500 µL of the serum-free medium to each chamber, the chamber was washed three times with the serum-free medium. Approximately 7 × 10^4^ cells in 300 µL serum-free DMEM were then placed in the upper chamber and 800 µL CM was added to the lower chamber. After 48 h, the cells on the upper surface of the membrane were removed and the membrane was treated with ~400 µL of 75% ethanol in a fresh 12-well plate for 40 min. The cells, which invaded the lower side of the membrane, were stained with Crystal Violet (diluted 1:25 in water) for 30 min. At least five randomly chosen images were taken with an inverted optical microscope (magnification, 100×, Nikon, Tokyo, Japan), and the average number of stained cells, which represented the invasion capacity, was determined.

The wound-healing scratch assay was utilized to evaluate two-dimensional cell migratory behavior [33]. Approximately 4 × 10^5^ cells were seeded in 12-well plates, and after the cell attachment, a scratch was made on the cell layer with a plastic pipette tip. Floating cells were removed by washing two times using a serum-free medium and CM was given. Images of the cell-free areas were captured at 0 h and 24 h after scratching via an inverted microscope with a magnification of 40×. The areas of eight images in each group were quantified with Image J.

### 2.6. Western Blot Analysis and Mass Spectrometry

Cells were lysed in a radio-immunoprecipitation assay buffer with protease inhibitors (PIA32963, Thermo Fisher Scientific, Waltham, MA, USA) and phosphatase inhibitors (2006643, Calbiochem, Billerica, MA, USA). After cell lysis, proteins were fractionated by 10–15% SDS gels and electro-transferred to polyvinylidene difluoride transfer membranes (IPVH00010, Millipore, Billerica, MA, USA). After blocking 1 h with a blocking buffer (1706404, Bio-Rad, Hercules, CA, USA), the membrane was incubated overnight with primary antibodies and then with secondary antibodies conjugated with horseradish peroxidase for 45 min (7074S/7076S, Cell Signaling, Danvers, MA, USA). We used antibodies against Runx2, Snail, Lrp5, β-catenin, TNFα, IL1β, GAPDH, p-P38, P38 (8486, 3879, 5731, 9562, 3707, 12242, 2118, 4511, 9212, respectively, Cell Signaling), NFATc1, cathepsin K, MMP13, L1CAM (sc-7294, sc-48353, sc-30073, sc-374046, respectively, Santa Cruz Biotechnology), Hsp90ab1 (ab203085, Abcam), and β-actin as a control (A5441, Sigma). Of note, we detected the cleaved MMP13 fragment rather than proMMP13. The level of proteins was determined using a SuperSignal west femto maximum sensitivity substrate (PI34096, Thermo Fisher Scientific), and a luminescent image analyzer (LAS-3000, Fuji Film, Tokyo, Japan) was used to quantify signal intensities [34]. All the whole Western blot figures can be found in Appendix A. Proteins from Immunoprecipitation were analyzed in the Dionex UltiMate 3000 RSLC nano-system combined with the Q-exactive high-field hybrid quadrupole orbitrap mass spectrometer (Thermo Fisher Scientific). Proteins were first digested on beads using trypsin/LysC as described previously except digestion was performed in 50 mM ammonium bicarbonate buffer instead of urea. Digested peptides were then desalted using mini spin C18 spin columns (1910-050, The Nest Group, Southborough, MA, USA) and separated using a trap and 50-cm analytical columns [35,36]. Raw data were processed using MaxQuant (Max Planck Institute of Biochemistry, Martinsried, Germany) against the Uniprot mouse protein database at a 1% false discovery rate allowing up to two missed cleavages. MS/MS counts were used for relative protein quantitation. Proteins identified with at least one unique peptide and 2 MS/MS counts were considered for the final analysis.

### 2.7. Immunoprecipitation

Immunoprecipitation was conducted with an immunoprecipitation starter pack kit (17600235, Cytiva, Marlborough, MA, USA), using the procedure the manufacturer provided. In brief, 20 µL of protein A sepharose was washed twice with PBS and incubated with 2 µg of antibodies for Hsp90ab1. In parallel, normal IgG was prepared for negative control. We employed C28/I2 chondrocytes lysate for Hsp90ab1. The antibody-cross-linked beads were incubated overnight with 600 µL protein samples on a shaker. The beads were collected by centrifugation, washed three times with PBS, and resuspended for Western blotting. The protein samples before the immunoprecipitation were used as positive controls. Western blotting was conducted using antibodies against GAPDH. The same procedure was employed to evaluate the interaction of GAPDH with L1CAM in SW1353 cells.

### 2.8. Plasmid Transfection and RNA Interference

Cells were transfected with plasmids encoding Lrp5, β-catenin, and GAPDH (#115907, #31785, #70148, Addgene, Watertown, MA, USA), while a blank plasmid vector (FLAG-HA-pcDNA3.1; Addgene) was used as a control [30]. siRNAs were employed for GAPDH and L1CAM (s103461, s536518, Thermo-Fisher) together with a nonspecific negative control siRNA (Silencer Select #1, Life Technologies; On-target Plus Non-targeting Pool, Dharmacon). Cells were transiently transfected with siRNA with Lipofectamine RNAiMAX (13778075, Life Technologies, Carlsbad, CA, USA).

### 2.9. Animal Models

The experimental procedures using animals were approved by the Indiana University Animal Care and Use Committee (SC330R, approved on 12 March 2021) and were complied with the Guiding Principles in the Care and Use of Animals endorsed by the American Physiological Society. Mice were housed five per cage and provided with mouse chow and water ad libitum. They were randomly assigned to the placebo and treatment groups. In the mouse model of osteolysis, ten NOD/SCID/γ(−/−) (NSG) female mice per group received an injection of SW1353 cells (2.5 × 10^5^ cells in 20 µL PBS) into the right tibia as an intra-tibial injection. YS49 CM was given daily as an intravenous injection to the tail vein. Mice were sacrificed in 14 days and the tibiae were harvested for µCT imaging and histology.

### 2.10. µ. CT Imaging and Histology

The tibia was harvested for µCT imaging using Skyscan 1172 (Bruker-MicroCT, Kontich, Belgium) and histology. Using the manufacturer-provided software, CT scans were performed with a pixel size of 8.99 μm and the captured images were reconstructed (nRecon v1.6.9.18) and analyzed (CTan v1.13). Using µCT images, trabecular bone parameters such as bone volume ratio (BV/TV), bone mineral density (BMD), and trabecular number (Tb.N) were determined in a blinded fashion. For histology, tibia samples were decalcified, dehydrated through a series of graded alcohols, cleared in xylene, and embedded in paraffin. To determine the distribution of tumor cells, we evaluated the sections from the proximal tibia at 60-μm intervals. Images were taken from five locations per slide, and the tumor area was quantified as a ratio of the tumor-colonized area to the total area in a blinded fashion.

### 2.11. Statistical Analysis

For cell-based experiments, three or four independent experiments were conducted and data were expressed as mean ± S.D. Statistical significance was evaluated using a one-way analysis of variance (ANOVA). Post hoc statistical comparisons with control groups were performed using Bonferroni correction with statistical significance at *p* < 0.05. In animal experiments, we employed 10 mice per group to obtain statistically significant differences in bone volume ratio as a primary outcome measure. The single and double asterisks in the figures indicate *p* < 0.05 and *p* < 0.01, respectively.

## 3. Results

### 3.1. Anti-Tumorigenic Effects of YS49-Treated SW1353/MSC-Derived CM

To activate PI3K/Akt signaling, SW1353 CS cells were treated with 50 µM of YS49, a pharmacological activator of PI3K signaling, for one day. The culture medium was refreshed to remove YS49 and CM was collected after 1 additional day of culture. We observed that YS49-treated SW1353 CS-derived CM (YS CS CM) reduced EdU-based proliferation in 2 days, scratch-based motility in 1 day, transwell invasion in 2 days, and MTT-based viability in 2 and 3 days of SW1353 cohort cells (Figure 1A–C, Appendix A). Besides CS cells as a source of iTSCs, we employed MSCs. YS49-treated MSC-derived CM (YS MSC CM) also reduced EdU-based proliferation of SW1353 CS cells in 2 days, scratch-based motility in 1 day, transwell invasion in 2 days, and MTT-based viability in 2 days (Figure 1D–F, Appendix A). Consistently, YS CM suppressed MTT-based viability in 2 days, EdU-based proliferation in 2 days, and transwell invasion in 2 days of OUMS-27 CS cells, the other CS cell line (Appendix A). Furthermore, YS MSC CM downregulated Runx2 and Snail in SW1353 CS cells (Appendix A). Collectively, CS cells and MSCs were converted to iTSCs and their CMs presented anti-tumor capabilities.

### 3.2. Promotion of Tumorigenic Responses by TNFα

We have so far shown that PI3K-activated CS/MSC-derived CM presents anti-tumor actions. We next evaluated the tumor-inflammation relationship using TNFα, a pro-inflammatory cytokine, as a probe. Before testing the effect of CM, we first examined the outcome for the direct application of TNFα to tumor cells. The incubation with 10 ng/mL of TNFα promoted MTT-based viability in 2 days, EdU-based proliferation in 2 days, and transwell invasion in 2 days of SW1353 CS cells (Figure 2A–C). The same stimulatory responses were observed with OUMS-27 CS cells (Appendix A), as well as 4T1.2 mammary tumor cells (Appendix A). Consistently, TNFα upregulated Runx2, a tumor-promoting transcription factor, and Snail, a regulator of EMT, in SW1353 CS cells (Appendix A). We then examined the effect of CMs derived from TNFα-treated CS cells, MSCs, and chondrocytes. Importantly, TNFα-treated CM reduced EdU-based proliferation of SW1353 CS cells in 2 days (Figure 2D–F) and scratch-based motility in 1 day (Appendix A). It also downregulated the expression of Runx2 and Snail in SW1353 CS cells (Appendix A). Furthermore, TNFα-treated MSC-derived CM reduced the MTT-based viability of OUMS-27 CS cells in 2 days, EdU-based proliferation in 2 days, and transwell invasion in 2 days (Appendix A). Taken together, the result revealed the differential outcome of CS cells in response to TNFα, depending on the direct or indirect CM-mediated application.

### 3.3. Generation of Anti-Inflammatory CM

We then evaluated the effects of TNFα on inflammatory responses. As expected, the administration of TNFα to C28/I2 chondrocytes, MSCs, and synovial cells elevated the levels of TNFα and MMP13 in TNFα-treated cells (Appendix A). However, the effect of TNFα via CM was the opposite. A series of chondrocyte-derived CMs, which were generated by culturing with 2 to 10 ng/mL of TNFα, downregulated TNFα and MMP13 in control chondrocytes (Figure 3A). Besides chondrocyte-derived CM, MSC-derived CM, as well as MH7a synovial cell-derived CM, were converted to tumor suppressive by the treatment with YS49, an activator of PI3K signaling. Their application reduced the level of TNFα and MMP13 in C28/I2 chondrocytes (Figure 3A). Taken together, the result revealed that anti-inflammatory CM can be derived from chondrocytes, MSC, and synovial cells by the culturing with TNFα and the activation of PI3K signaling (Figure 3B).

### 3.4. Context-Dependent Effects by the Activation of PI3K/Akt Signaling

In response to the direct application of YS49, we observed the elevated levels of TNFα and MMP13 in SW1353 CS cells, C28/I2 chondrocytes, MSCs, and MH7a synovial cells (Appendix A). By contrast, culturing in a group of YS49-treated CMs, derived from SW1353 CS cells, C28/I2 chondrocytes, MSCs, and synovial cells, downregulated TNFα and MMP13 in C28/I2 chondrocytes (Figure 4A). Collectively, the results so far revealed that anti-inflammatory CM can be generated by the application of YS49 as well as TNFα (Figure 4B).

Besides the activation of PI3K signaling, we examined the effect of Wnt activation on the inflammatory responses. In C28 chondrocytes, β-catenin and Lrp5, a Wnt co-receptor, were overexpressed and their CMs were harvested. We observed that both β-catenin- and Lrp5-overexpressing chondrocyte-derived CM reduced the levels of TNFα and MMP13 in C28 control chondrocytes (Appendix A). While we have shown that iTSCs can be generated by the activation of PI3K and Wnt pathways (16,17), the results in this study also indicated that it is possible to generate iISCs by activating these pathways.

### 3.5. Beneficial Role of Extracellular Hsp90ab1

To access the mechanism of anti-tumor, anti-inflammatory actions of YS MSC CM, we selected five tumor suppressors (Hsp90ab1, calreticulin, histone H4, polyubiquitin C, and enolase 1) based on our previous whole-genome proteomics analysis [17,19,23]. Among them, Hsp90ab1, a heat shock protein acting as a molecular chaperon, showed the most pronounced anti-inflammatory effect in C28 chondrocytes by downregulating TNFα and MMP13 (Appendix A). Furthermore, the application of 800 ng/mL of recombinant Hsp90ab1 proteins significantly elevated the ELISA-based level of aggrecan and type II collagen in C28/I2 chondrocytes in 3 days (Figure 5A). Thus, we further examined Hsp90ab1-mediated anti-tumor, anti-inflammatory actions.

### 3.6. Interaction of Hsp90ab1 with Extracellular GAPDH

In immunoprecipitated proteins using an antibody against Hsp90ab1, mass spectrometry-based proteomics identified 1597 proteins in tumor cells and their ECM. Out of 1597 proteins, a relative abundance score of 39 proteins was above 20 (Appendix A) and a shortlist of 10 proteins had a score above 44 (Figure 5B). Among six non-heat shock proteins in the shortlist, Western blotting confirmed the coprecipitation of vimentin and glyceraldehyde 3-phosphate dehydrogenase (GAPDH) with Hsp90ab1 (Figure 5C). Importantly, the incubation of SW1353 CS cells with 2 µg/mL of recombinant GAPDH proteins reduced MTT-based viability in 2 days, EdU-based proliferation in 2 days, and transwell invasion in 1 day (Figure 5D–F). We also evaluated the dose responses of SW1353 CS cells to the separate and combined application of GAPDH and Hsp90ab1 recombinant proteins. The result showed the additive and dose-dependent responses (Appendix A). However, no detectable effect was observed with recombinant vimentin proteins. Besides SW1353 CS cells, GAPDH reduced MTT-based viability of OUMS-27 CS cells in 2 days, EdU-based proliferation in 2 days, and scratch-based motility in 1 day (Figure 5G–I).

### 3.7. GAPDH as an Anti-Inflammatory Protein

Notably, we observed that the application of 2 µg/mL GAPDH proteins downregulated TNFα and MMP13 in C28 chondrocytes (Appendix A). Although GAPDH is primarily considered an intracellular enzyme for glycolysis, the result herein supported the anti-tumor and anti-inflammatory action of extracellular GAPDH. We employed RNA interference and overexpression of GAPDH and Hsp90ab1 and analyzed their role in anti-inflammatory actions (Figure 6A). The data also revealed that GAPDH was upregulated in response to Hsp90ab1, which was downregulated by silencing GAPDH in C28/I2 chondrocytes (Figure 6B). Consistently, silencing Hsp90ab1 in C28/I2 chondrocytes elevated the levels of p-p38 and TNFα, while the application of Hsp90ab1 reduced their levels (Figure 6C). We observed that silencing GAPDH in C28/I2 chondrocytes also elevated p-p38 and TNFα, while GAPDH-overexpressing C28/I2 chondrocytes reduced them (Figure 6C). In summary, extracellular and intracellular GAPDH acted as the anti-inflammatory agent to C28 chondrocytes by inhibiting p-p38 and TNFα.

### 3.8. GAPDH as an Anti-Tumor Protein and Its Interaction with L1CAM

So far, we examined the anti-inflammatory and anti-tumor actions of extracellular GAPDH. We next evaluated the role of intracellular GAPDH in SW1353 CS cells. Silencing GAPDH suppressed MTT-based viability in 2 days, transwell invasion in 2 days, and downregulated TNFα and MMP13 (Figure 6D–F). By contrast, GAPDH-overexpressing SW1353 CS cells presented the elevation of viability and invasion, together with the elevated levels of TNFα and MMP13 (Figure 6D–F). Of note, GAPDH is reported to interact with L1CAM, a transmembrane protein with an oncogenic capability [37]. Western blotting confirmed that GAPDH immunoprecipitated with L1CAM in SW1353 CS protein extracts (Figure 6G). Notably, the tumor-suppressive effect of extracellular GAPDH was suppressed by silencing L1CAM in SW1353 CS cells, indicating that L1CAM in tumor cells mediated GAPDH-driven anti-tumor action (Figure 6H–J).

### 3.9. Suppression of Osteoclast Development and Tumor-Driven Bone Loss by YS MSC CM

We also examined the effect of YS MSC CM on the development of osteoclasts in vitro and tumor-driven bone loss in vivo. First, we observed that YS MSC CM suppressed the differentiation of RANKL-stimulated RAW264.7 pre-osteoclasts into multinucleated osteoclasts in 5 days, and RAW264 cells reduced NFATc1, a transcription factor for osteoclastogenesis, and cathepsin K, a potent proteinase for bone resorption (Figure 7A,B). Consistent with the in vitro action of YS MSC CM, NSG mice, which were inoculated with SW1353 CS cells in the tibia, exhibited a reduction in bone loss by administrating YS MSC CM in 14 days. The daily i.v. administration of YS MSC CM elevated the bone volume ratio, bone mineral density, and trabecular number in the CS-invaded proximal tibia (Figure 7C,D). Moreover, the result revealed that the administration of YS MSC CM reduced the levels of TNFα, IL1β, and MMP13 in the synovium and proximal tibia of NSG mice (Appendix A).

## 4. Discussion

This study presented that the chondroprotective proteome can be generated from MSCs and chondrocytes by activating PI3K signaling with YS49 and treating them with TNFα. When switched on, both PI3K and TNFα signalings acted tumorigenic in CS cells and inflammatory in chondrocytes. However, YS49 and TNFα converted skeletal host cells such as MSCs and chondrocytes into iTSCs and iISCs, and they generated tumor-suppressive and inflammation-suppressive CMs. We observed that YS49 and TNFα-treated CMs inhibited the proliferation, migration, and invasion of CS cells. Those CMs downregulated Runx2 and MMP9 in CS cells, as well as NFATc1 and cathepsin K in osteoclasts, whereas they decreased TNFα and MMP13 in chondrocytes (Figure 8). CM also upregulated ECM proteins such as aggrecan and type II collagen in chondrocytes. Collectively, using a counterintuitive approach with the PI3K activator and pro-inflammatory cytokine, this study presented the possibility of developing a novel MSC-derived CM-based option for the treatment of CS and inflammatory arthritis.

In analogous to cell competitions during the development of Drosophila and mouse embryos, we have shown that PI3K signaling can become both pro and anti-oncogenic to OS, depending on the intracellular and extracellular domains. Numerous studies have been directed to develop efficient inhibitors of PI3K signaling since its aberrant activation can promote the initiation and metastasis of various cancers [38]. By contrast, we have shown in this study that PI3K-activated MSCs can produce anti-tumor secretory proteomes and prevent tumor-induced osteolysis by inhibiting the progression of CS cells and the differentiation of bone-resorbing osteoclasts. Consistently, we have previously shown by using human lymphocytes that a pharmacological PKA inhibitor generated pro-tumorigenic CM, while a PKA inhibitor produced anti-tumorigenic CM [39]. In this study, we extended this paradoxical strategy for the generation of tumor-suppressive CM and produced inflammation-suppressive CM by activating pro-inflammatory signaling. 

GAPDH is a glycolytic enzyme for the production of ATP and its high expression in cancer cells is associated with poor prognosis. It was thus unexpected that GAPDH, which was immunoprecipitated with Hsp90ab1, acted as a tumor suppressor as well as an anti-inflammatory protein [40], although the participation of GAPDH in diverse processes including pro-apoptotic and anti-inflammatory responses has been reported [41,42,43]. GAPDH is reported to maintain intact mitochondria and facilitate tumor survival and chemotherapeutic resistance [44], and its deregulated expression assists in the vascularization and aggressiveness of lymphoma [45]. However, it is also reported that GAPDH induces an arrest of telomere maintenance and the proliferation of cancer cells [46]. While Hsp90ab1 may assist the stability of GAPDH as a molecular chaperon, we observed that they both served as tumor suppressors to CS cells. p38 and TNFα are both pro-inflammatory cytokines [47,48]. Extracellular Hsp90ab1 elevated the level of GAPDH in C28 chondrocytes and reduced p-p38 and TNFα. Consistently, silencing Hsp90ab1 in C28 chondrocytes elevated TNFα, and overexpressing GAPDH in C28 chondrocytes reduced p-p38 and TNFα. Extracellular GAPDH inhibited the viability and proliferation of SW1353 and OUMS27 CS cells via the interaction with L1CAM and downregulated TNFα and MMP13 in C28 chondrocytes. Taken together, GAPDH was identified as a critical player in the intracellular and extracellular domains for suppressing tumor progression and inflammatory responses.

From a translational viewpoint, MSCs are considered a feasible source of iTSCs and iISCs since they can be harvested from the bone marrow of a patient with CS or inflammatory arthritis. A patient may receive autologous MSC-derived CM or the implantation of iTSCs and iISCs. The current ClinicalTrials.gov site lists over 200 MSC-linked studies for cancer treatment and 35 studies for the treatment of arthritis in the U.S., from various sources including adipose tissue MSC. It is important to evaluate potential side effects associated with the administration of MSC-derived CM. Compared to the placebo that did not show a detectable weight increase during the tumor-bearing experiment, the weight increase was observed in the MSC CM-injected group. Although indirect, the result does not indicate CM’s toxicity, specifically, induction of appetite loss, since mice tend to lose body weight in response to the treatment with chemotherapeutic agents. Furthermore, the inhibitory effect of iTSC CM is reported selective for tumor cells rather than non-tumor cells [19]. Besides MSCs, the use of lymphocytes and mononuclear cells from patient-derived or peripheral blood might also be a proper source of host cells for generating iTSCs and iITSCs.

This study only focused on secretory proteomes, although a whole secretome contains other factors such as nucleic acids, lipids, metabolites, and neurotransmitters. Of note, the cell culture conditions for harvesting CM should have effects on the levels of varying proteins in CM. We observed that the treatment of CMs with nucleases, ultracentrifugation to remove micro and nanoparticles such as exosomes, and filtering with a 3-kD filter did not significantly alter the joint-protective action of CM. Besides PI3K signaling and TNFα, other signaling such as Wnt signaling and cytokines such as IL1 may generate iTSCs and iISCs. Further studies are required to identify the most effective procedure for generating tumor-suppressive, inflammation-suppressive CM for MSCs. It is also necessary to conduct preclinical studies for evaluating the potential benefits and adverse effects of these strategies, if any, using animal models for OC and OA. Besides MMP9, MMP13, aggrecan, and type II collagen, the action of CMs on other MMPs, ADAMTS, TIMPS, and other ECM proteins should be evaluated for the protection of bone and joint tissues.

## 5. Conclusions

We demonstrated that MSCs and chondrocytes can generate the chondroprotective proteome by the activation of PI3K and incubation with TNFα. Furthermore, extracellular Hsp90ab1 was enriched in the proteome, and it induced anti-tumor, anti-inflammatory actions via the GAPDH-L1CAM regulatory axis. The unconventional approach in this study was based on the context-dependent role of PI3K signaling, as well as TNFα, Hsp90ab1, and GAPDH. The chondroprotective mechanism was likely linked to the competition among cells, which has been observed in embryonic development as well as the interactions among heterogeneous tumor populations. In summary, the results of this study are expected to contribute to developing an unconventional option for the treatment of CS and inflammatory arthritis by taking advantage of tumorigenic and inflammatory signaling.

## Figures and Tables

**Figure 1 cancers-14-03039-f001:**
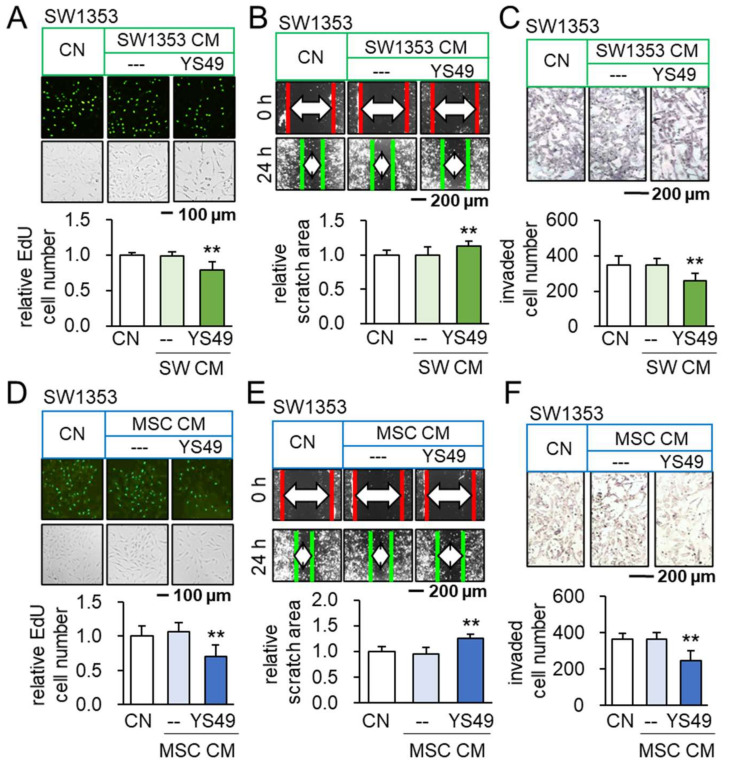
Anti-tumorigenic effects of YS49-treated SW1353/MSC-derived conditioned medium. CN = control, and CM = conditioned medium. The concentration of YS49 was 50 µM. The double asterisk indicates *p* < 0.01. (**A**–**C**) Reduction in EdU-based proliferation, scratch-based motility and transwell invasion of SW1353 CS cells by YS49-treated CS-derived CM (*n* = 6). (**D**–**F**) Reduction in EdU-based proliferation, scratch-based motility and transwell invasion of SW1353 CS cells by YS49-treated MSC-derived CM (*n* = 6).

**Figure 2 cancers-14-03039-f002:**
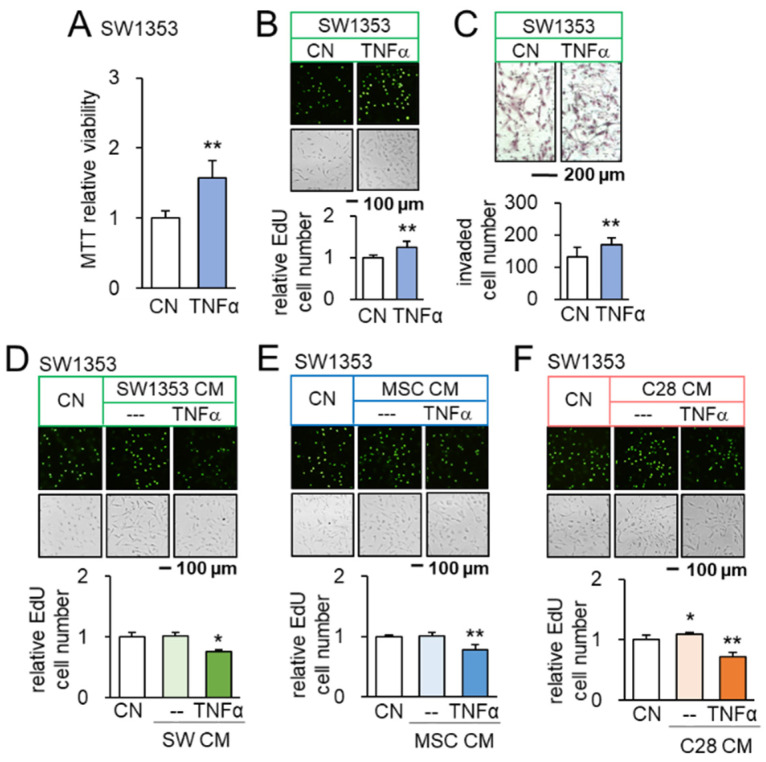
Opposing effects of TNFα and TNFα-treated CM on SW1353 CS cells. CN = control, CM = conditioned medium, and C28 = C28/I2 chondrocytes. The single and double asterisks indicate *p* < 0.05 and 0.01, respectively. (**A**–**C**) Increase in MTT-based viability, EdU-based proliferation, and transwell invasion of SW1353 cells in response to 10 ng/mL of TNFα (*n* = 6). (**D**) Upregulation of Runx2 and Snail in SW1353 cells by 10 ng/mL of TNFα (*n* = 6). (**E**,**F**) Reduction in EdU-based proliferation of SW1353 cells by TNFα-treated CS cells, MSC, and C28 chondrocyte-derived CMs, respectively (*n* = 6). The concentration of TNFα was 10 ng/mL.

**Figure 3 cancers-14-03039-f003:**
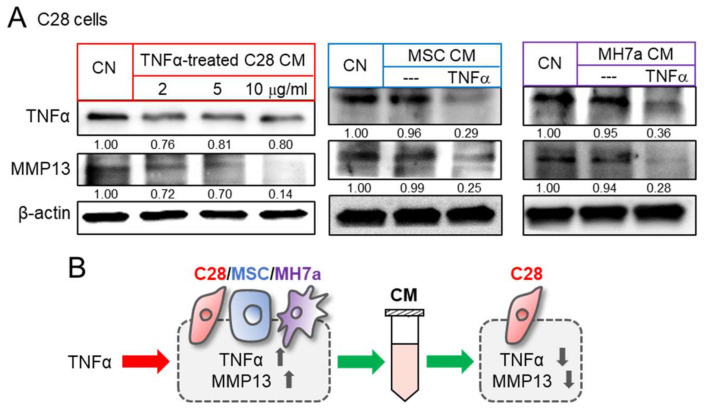
Reduction in TNFα and MMP13 in C28 chondrocytes by TNFα-treated CM. CN = control, C28 = C28/I2 chondrocytes, and MH7a = MH7a synovial cells. (**A**) Downregulation of TNFα and MMP13 in C28/I2 chondrocytes by TNFα-treated CS cell, MSC, and C28 chondrocyte-derived CMs, respectively. MSCs and MH7a cells were treated with 10 ng/mL of TNFα. (**B**) Schematic diagram for the effects of TNFα and TNFα treated cells-derived CM. Full Western blot images are available in Appendix A.

**Figure 4 cancers-14-03039-f004:**
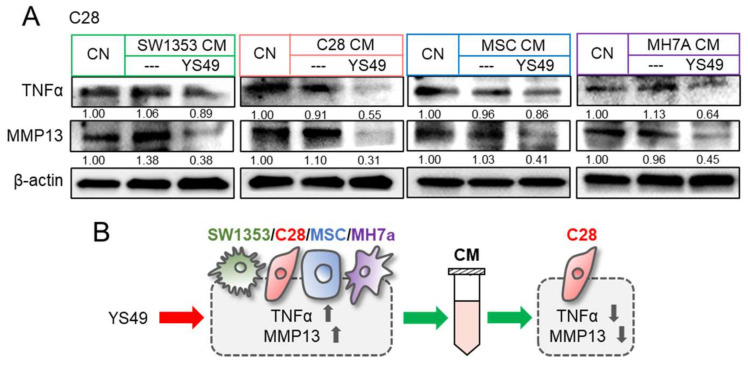
Reduction in TNFα and MMP13 by YS49 CM. CN = control, CM = conditioned medium, YS = YS49, SW = SW1353 CS cells, C28 = C28/I2 chondrocytes, and MH7a = MH7a synovial cells. (**A**) Downregulation of TNFα and MMP13 in C28/I2 chondrocytes by YS49-treated SW1353 CS cell, C28 chondrocyte, MSC, and MH7a synovial cell-derived CMs, respectively. The concentration of YS49 was 50 µM. (**B**) Schematic diagram for the effects of YS49 and YS49 treated cells-derived conditioned medium. Full Western blot images are available in Appendix A.

**Figure 5 cancers-14-03039-f005:**
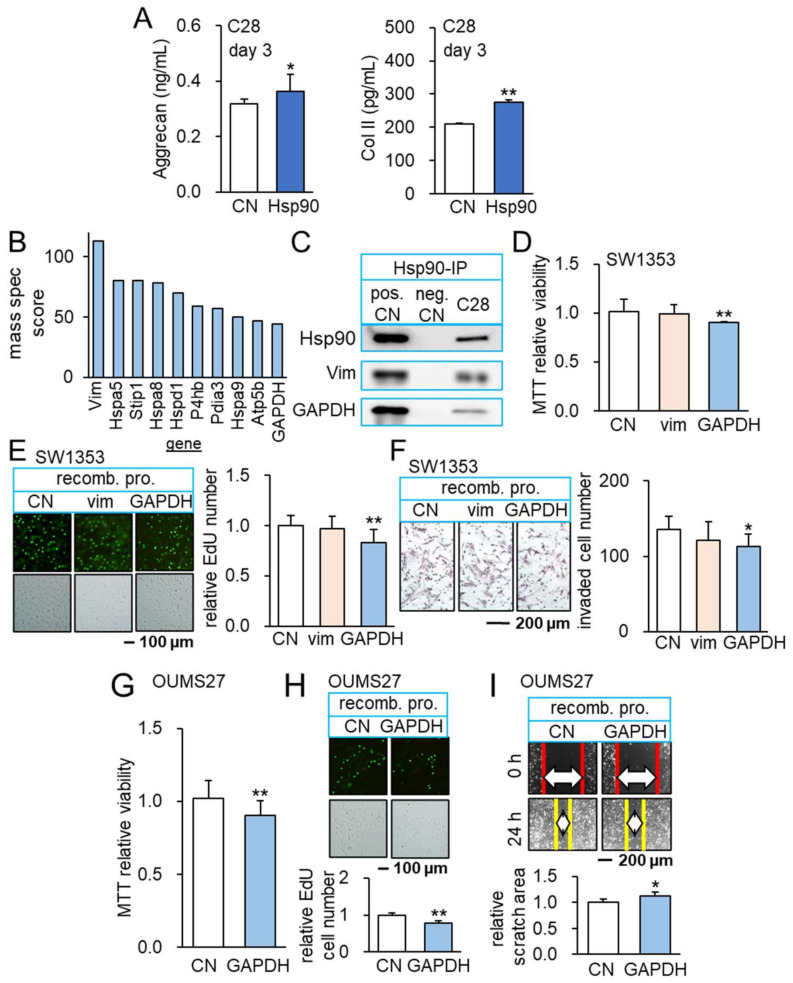
Anti-inflammatory role of Hsp90ab1 and GAPDH. The single and double asterisks indicate *p* < 0.05 and 0.01, respectively. (**A**) Elevation of aggrecan and type II collagen in C28/I2 chondrocytes in response to 1 µg/mL of Hsp90ab1 (*n* = 6). (**B**) Co-immunoprecipitated proteins with Hsp90ab1. (**C**) Co-immunoprecipitated GAPDH with Hsp90ab1 in C28/I2 chondrocytes. (**D**–**F**) Reduction in MTT-based viability, EdU-based proliferation, and scratch-based motility of SW1353 cells in response to 2 ug/mL of GAPDH (*n* = 6). (**G**–**I**) Reduction in MTT-based viability, EdU-based proliferation, and transwell invasion of OUMS-27 cells in response to 2 µg/mL of GAPDH (*n* = 6). Full Western blot images are available in Appendix A.

**Figure 6 cancers-14-03039-f006:**
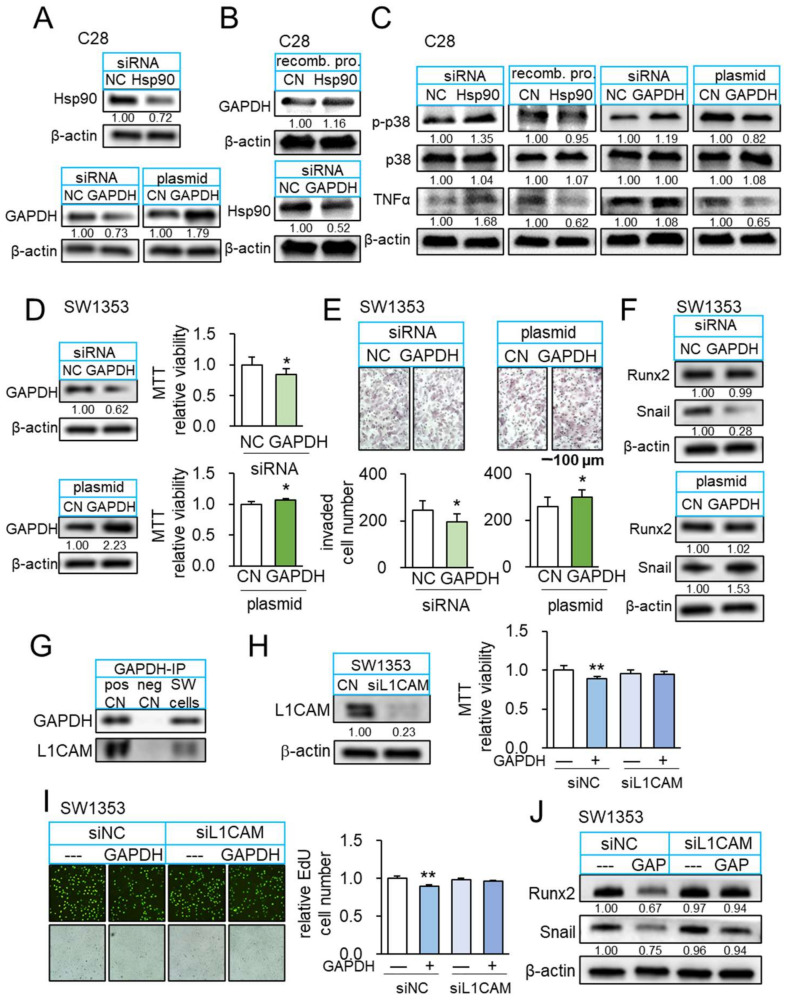
Interaction of GAPDH and L1CAM. The single and double asterisks indicate *p* < 0.05 and 0.01, respectively. (**A**) Expression levels of Hsp90ab1 and GAPDH after their silencing or overexpression in C28/I2 chondrocytes. (**B**) Upregulation of GAPDH in response to recombinant Hsp90ab1proteins, and downregulation of Hsp90ab1 by silencing GAPDH in C28/I2 chondrocytes. (**C**) Elevation of p-p38 and TNFα by silencing Hsp90ab1 in C28/I2 chondrocytes, while their reduction in response to recombinant Hsp90ab1 protein. Elevation of p-p38 and TNFα by silencing GAPDH in C28/I2 chondrocytes, while their reduction in GAPDH-overexpressing C28/I2 chondrocytes. (**D**,**E**) Reduction in MTT-based viability and transwell invasion by silencing GAPDH in SW1353 cells and the opposite effect in GAPDH overexpressed SW1353 cells (*n* = 6). (**F**) Downregulation of Snail by silencing GAPDH in SW1353 cells and elevation of Snail in GAPDH overexpressed SW1353 cells. (**G**) Co-immunoprecipitated L1CAM with GAPDH. (**H**,**I**) Suppression of GAPDH-mediated inhibition of MTT-based viability and EdU-based proliferation of SW1353 cells by RNA silencing of L1CAM (*n* = 6). (**J**) Downregulation of Runx2 and Snail in response to recombinant GAPDH in SW1353 cells, and its suppression by silencing L1CAM. Full Western blot images are available in Appendix A.

**Figure 7 cancers-14-03039-f007:**
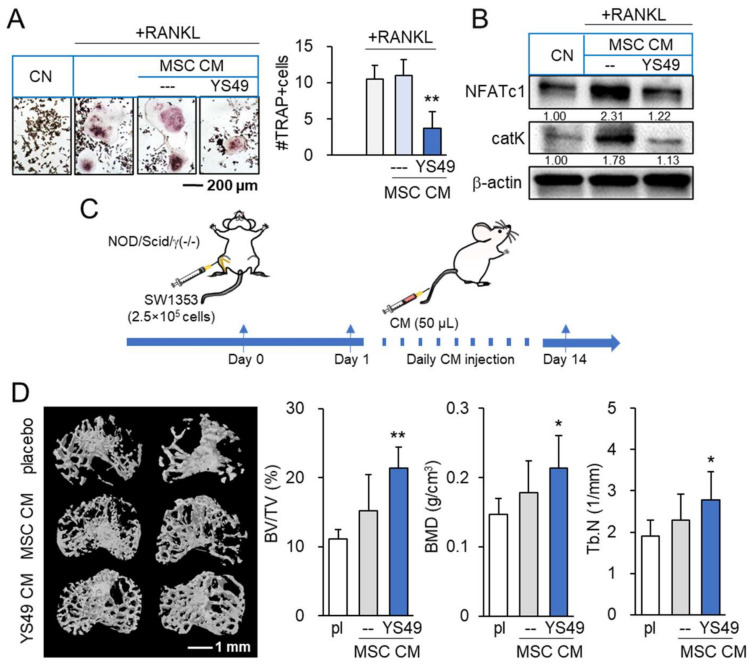
Suppression of osteoclast development and CS-driven bone loss by YS49-treated MSC-derived conditioned medium. pl = placebo, CM = conditioned medium, and YS49 CM = YS49-treated MSC-derived CM. The concentration of YS49 was 50 µM. The single and double asterisk indicate *p* < 0.05 and 0.01, respectively. (**A**,**B**) Suppression of osteoclast development and downregulation of NFATc1 and cathepsin K by YS49 CM (*n* = 6). (**C**) Timeline of the animal experiment. SW1353 cells were injected into the tibiae of NOD/SCID/γ (−/−) (NSG) female mice (8 mice per group). The administration of CM began on day 1 after the injection. YS49 CM was given daily as an intravenous injection to the tail vein. Mice were sacrificed in 14 days. (**D**) Reduction in trabecular bone loss in the CS-invaded proximal tibia by YS49 CM. BV/TV = bone volume ratio, BMD = bone mineral density, Tb.N = trabecular number, and Tb.Sp = trabecular separation (*n* = 8). Full Western blot images are available in Appendix A.

**Figure 8 cancers-14-03039-f008:**
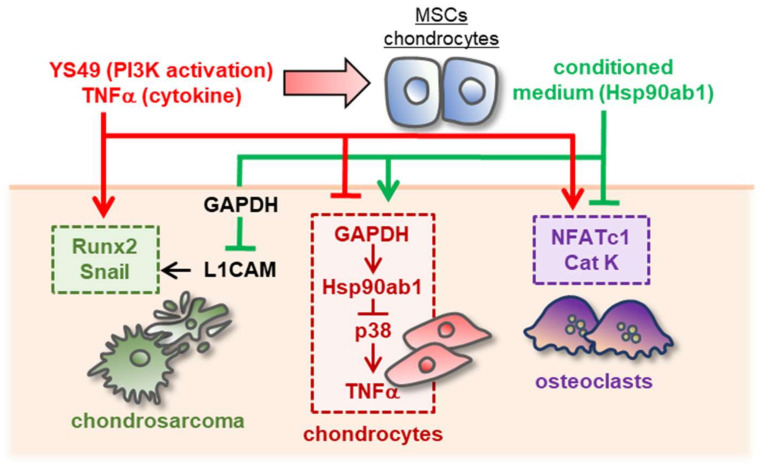
Schematic diagram for the putative regulatory mechanism with MSC/chondrocyte-derived CM. The activation of PI3K signaling or the administration of TNFα generated the chondroprotective MSC/chondrocyte-derived CM. The CM suppressed tumor progression via the GAPDH-L1CAM axis and downregulated TNFα by inhibiting p38 signaling. It also blocked the differentiation of bone-resorptive osteoclasts by downregulating NFATc1 and cathepsin K. By contrast, the activation of PI3K or the administration of TNFα to CS and chondrocytes stimulated tumorigenic and inflammatory responses.

## Data Availability

The data presented in this study are available in this article and the Appendix A.

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
