# Peer review of "Generation of the Chondroprotective Proteomes by Activating PI3K and TNFα Signaling"

_cancers, 2022, doi:10.3390/cancers14133039_

Round 1
Reviewer 1 Report
Sun et al describe an interesting and new strategy for the treatment of Chondrosarcoma and inflammatory arthritis. While the overall study is well-done, there are a few points that need to be addressed.
The biggest concern is the role of GAPDH in tumorigenic behaviors examined in this study. First of all, only a few reports linked GAPDH to potential oncogenesis, which still needs to be proved by additional studies. Furthermore, the malignant phenotypes and oncogenes affected by modulation of GAPHD were not dramatic (Figure 5 and 6). This makes it hard to believe that the proposed mechanism is mediated via GAPDH. More convincing data needs to be provided.
Other minor concerns: If possible, the reciprocal IPs should be done for Figures 5C and 6F. The MTT assays should be read daily, not just at day 2. Wells should be read on Day 0, Day 1, Day 2, and Day 3 and graphed accordingly. The mouse studies in Figure 7 should have some form of measurement for toxicity and potential side-effects of the proposed treatments. In the discussion, please discuss what could be some issues/downfalls with promoting the P13K pathway and if there is a danger of inducing other cancers.
Reviewer 2 Report
Major comments:
Regarding conditioned media: since the effects of matrix proteins from culture supernatants are being the levels of these proteins present in the conditioned media should be presented (perhaps by TCA precipitation, followed by western blots). This will indicate the relative amounts of each of the matrix proteins needed to see the effects observed.
Figure 2. Performing the TNFa treatments and/or TNFa conditioned medium treatments should be done side-by-side, with relevant controls. This would enable us to gauge the effects of TNFa vs conditioned media and present the differences in one panel.
Figure 3: Western blot for anti-MMP13: Please indicate the state of ‘Pro’ and ‘active’ forms of MMP13 (for example, as detected by MAB913, R&D Systems).
3A: For the panels MSC CM and MH7a CM, please indicate the amount of TNFa used in the figure legend.
Figure 4: (Same as Fig. 3A) Western blot for anti-MMP13: Please indicate the state of Pro and active forms of MMP13 (for example, as detected by MAB913, R&D Systems).
Fig. 5C: Perhaps outside the scope of this study; but using recombinant HSP90ab1, Vimentin and GAPDH proteins, in vitro pulldowns could be performed to tease out the nature of their interactions with HSP90ab
Figure 6:
6A, 6B, 6E and 6J: Please include a western blot to show the expression levels of the proteins being perturbed or over expressed (western blot with antibody to the tag?) in the respective panels (as was done in Fig. 6G: si: L1CAM; WB: L1CAM).
Figure 7: Please indicate the number of mice in each group in the figure legend.
Minor revisions:
-
Fig 3A: The labels could be made uniform. Instead of “TNFa treated C28 CM” “MSC CM” “MH7a CM”, probably C28CM, MSC CM, MH7a CM would suffice?
-
Line 85: Please cite the primary literature article as reference. Reference no. 13 is a review article.
-
Line 87: Claveria et. al. (Reference 14) show that heterogenous Myc levels in epiblast cells result in a self-selection of cells with higher expression. The wording “...cells are reported to remove neighboring cells with a low level of c-Myc” is at the very least gross misinterpretation of Claveria et. al.. In either case, expression of various proteins or transcription factors can be simplified and explained as “survival of the fittest” cell; not a win-lose scenario.
-
Line 140-142: Please clarify the language so as to indicate if/which cells were treated YS49 or NBP2 (TNFa) or both YS49+NBP2.
-
Lines 156-159: Minor concern/clarification: The process described in “2.3. Preparation of conditioned medium (CM) and ELISA assay” sounds to be too stringent (overnight spin at 100,000g). Is getting rid of exosomes critical to have the ‘protective’ effects of conditioned media described herein, while simultaneously avoiding tumorigenic effects concomitant with PI3K activation? (This reviewer is not well versed with the intricate nuances in generating conditioned media.)
-
Please adhere to one form of writing: MH7A (or MH7a).
-
For all the reagents used, please include the relevant catalog numbers, especially the antibodies used. For the most part the catalog numbers are included; please add the info for the reagents where it is missing (lines 211-213). This would be a great help to the community at large.
-
Figure panels involving statistics: Please list n (=3 or 4, as mentioned in the methods section) in the figure legends as well. An explanation as to repeats and how/what statistical tests were performed should be better explained in the legend and relevant text.
Round 2
Reviewer 1 Report
The authors improved the manuscript. I do not have any concerns.